# Examining the relationship between diarrhea and linear growth in Kenyan HIV-exposed, uninfected infants

Emily L. Deichsel [1¤]*, Grace C. John-Stewart[1,2,3,4], Judd L. Walson[1,2,3,4,5], Dorothy Mbori-Ngacha[6], Barbra A. Richardson[2,7], Brandon L. Guthrie[1,2], Carey Farquhar[1,2,4], Rose Bosire[8], Patricia B. Pavlinac[2]

1 Department of Epidemiology, University of Washington, Seattle, Washington, United States of America, 2 Department of Global Health, University of Washington, Seattle, Washington, United States of America, 3 Department of Pediatrics, University of Washington, Seattle, Washington, United States of America, 4 Department of Medicine, University of Washington, Seattle, Washington, United States of America, 5 Child Acute Illness and Nutrition (CHAIN) Network, Nairobi, Kenya, 6 United Nations Children's Fund (UNICEF), New York, New York, United States of America, 7 Department of Biostatistics, University of Washington, Seattle, Washington, United States of America, 8 Center for Public Health Research, Kenya Medical Research Institute (KEMRI), Nairobi, Kenya

¤ Current address: Center for Vaccine Development and Global Health, University of Maryland School of Medicine, Baltimore, Maryland, United States of America
* edeichsel@som.umaryland.edu

**Data Availability Statement:** Data cannot be shared publicly due to patient privacy and ethical approvals received. However, data will be available from the University of Washington Global Center

## Abstract

### Background

Diarrhea in infancy can compromise linear growth and this relationship is likely influenced by diarrhea severity, number of episodes, and the timing of those episodes. HIV exposed, uninfected infants (HEU) have higher risk of growth faltering, infectious morbidity and mortality than HIV-unexposed infants and may be representative of children particularly vulnerable to diarrhea-associated linear growth faltering.

### Methodology/Principal findings

We utilized data from a cohort of Kenyan HEU infants followed from birth to 12 months of age. Infant length and morbidity were ascertained at monthly study visits and sick visits. Longitudinal models estimated the association between diarrhea severity and length-for-age Z-score (LAZ) in the following month, at 12 months of age, and in 6-month intervals. The 372 enrolled infants experienced an average of 2.15 episodes (range: 0–8) of diarrhea and 0.54 episodes (0–4) of moderate-to-severe diarrhea (MSD) between birth and 12 months. Surviving infants had a mean LAZ of -0.97 (standard deviation: 1.2) at 12 months. MSD was significantly associated with an average loss of 0.14 (95% Confidence Interval [CI]: -0.24, -0.05, p = 0.003) in LAZ one month after the episode. Linear growth outcomes were not predicted by cumulative episodes of diarrhea, or timing of diarrhea during infancy.

for Integrated Health of Women, Adolescence, and Children for researchers who meet the criteria for access to confidential data. Please direct requests to info@globalwach.org.

**Funding:** This work was supported by Eunice Kennedy Shriver National Institute of Child Health and Human Development of the National Institutes of Health [grant numbers R01 HD-23412 to GJS, 1F31 HD-089507 to ELD]. The funder had no role in the study design, data collection, and analysis, decision to publish, or preparation of the manuscript. https://www.nih.gov/about-nih/what-we-do/nih-almanac/eunice-kennedy-shriver-national-institute-child-health-human-development-nichd.

**Competing interests:** The authors have declared that no competing interest exist.

## Conclusions/Significance

Diarrhea severity influenced the relationship between diarrhea and subsequent linear growth. HEU infants with MSD may benefit from nutritional interventions following severe diarrhea to protect against linear growth faltering.

## Introduction

Poor linear growth is a ubiquitous marker of chronic undernutrition. Stunting, defined as less than two deviations below the WHO length-for-age standard, is common in sub-Saharan Africa affecting more than a third of children under 5 [1]. Suboptimal linear growth in the first two years of life is associated with increased risk of mortality from infectious diseases [2], cognitive delays [3,4], and reduced adult work capacity [5].

Diarrheal disease has long been investigated as a potential cause of stunting with evidence that diarrhea leads to weight loss and eventually linear growth faltering in the absence of sufficient illness-free periods with adequate nutrition to support catch-up growth [6–11]. Results from a robust cohort study found limited effect of childhood diarrhea on linear growth [12,13] and interventional studies testing treatment and prevention of diarrhea to improve linear growth have had mixed results [14–17]. We hypothesize diarrhea severity [8], cumulative burden [18,19], and timing of diarrhea [11,20] may explain differences in the diarrhea and linear growth relationship seen in observational and interventional studies. Understanding these sources of heterogeneity may provide insight into how to target interventions to prevent linear growth faltering, and ultimately stunting, in young children.

In sub-Saharan Africa, children born to women living with HIV (WLWH), but whom themselves are uninfected (HEU) represent a growing population at risk for linear growth faltering. HEU children are at increased risk for infection with enteric pathogens, experience more frequent bouts of diarrhea, and are at higher risk of developing more severe diarrhea than their unexposed counterparts [21–24]. These vulnerabilities may be a consequence of in-utero HIV or antiretroviral treatment (ART) exposure, increased postnatal pathogen exposure from their HIV-infected mother [22,25,26], or sociodemographic challenges [22,24,27,28]. This population may serve as a model for discerning relationships between diarrheal illness and linear growth faltering because they are at high risk of both conditions. Utilizing data from a birth cohort of HEU infants from Nairobi, Kenya, we aimed to determine the effect of diarrhea severity, cumulative burden, and timing on linear growth.

## Methods

### Study design

This secondary analysis utilized data collected from a previously accrued cohort of HIV-infected women and their infants described in detail elsewhere [29–31]. In brief, from 1999–2002 HIV-infected women recruited during pregnancy (≤32 weeks gestation) in Nairobi, Kenya were followed with their infants for one year after birth. Mothers received short-course zidovudine to prevent maternal-to-child transmission of HIV according to Kenyan National guidelines at the time of the study [32]. In addition, women with severe immunosuppression (CD4 count <200 cells/μl) received cotrimoxazole prophylaxis. Mothers did not receive antiretroviral therapy (ART) during breastfeeding and infants did not receive cotrimoxazole prophylaxis, per the standard of care at the time of the study.

## Data collection

All data were collected following standardized procedures and data collection forms by study staff as part of the parent study. Infant recumbent length measurements were collected using a length board during monthly routine study visits from birth to 12 months of age. Length-for-age z-scores (LAZ) were calculated using the WHO Anthro macro developed for Stata and based on the 2006 WHO Child Growth Standard [33]. Infant morbidities, including diarrhea, were documented at monthly routine study visits through a clinical exam or when mothers were asked about infant illness and breastfeeding in the past month. Mothers were encouraged to bring their infant to the study clinic at any point during follow up if the child was sick, where morbidity diagnoses, including diarrhea, were recorded.

## Statistical analysis

Singleton or firstborn twin infants with documentation of sex, at least one negative HIV DNA polymerase chain reaction (PCR) test, and at least two recorded length measurements were included in the present analysis. Infants with a positive HIV DNA PCR test at or before one month of age were considered perinatally infected and excluded from the analysis. Infants were censored from the analysis cohort at their last HIV-negative test during the 12-month follow-up.

Diarrhea was defined as any diarrhea episode reported by the mother since the last study visit or a clinician diagnosis of diarrhea at any study visit (routine or sick visit). Diarrheal episodes occurring within 14 days of one another were counted as the same episode to conservatively avoid double counting of diarrhea episodes. Clinician diagnosis of diarrhea at sick visits was considered diarrhea even if diarrhea was not the primary diagnosis. An episode was classified as moderate-to-severe diarrhea (MSD) when diarrhea occurred with dysentery or dehydration, or there was a diarrhea associated hospitalization [20].

An infant was considered breastfed if the mother reported the infant receiving any breastmilk in the last 24 hours at a routine study visit. Incidence of diarrhea and average breastfeeding duration were calculated using interval censoring methods to censor person-time for missed visits and 7 days before and after a diarrhea episode (for incidence of diarrhea only).

We determined the effect of any diarrheal episode in a month (short-term effect), cumulative burden, and timing of a diarrheal episode on LAZ in the following month, throughout the first year of an infant's life. To determine the specific relationship of diarrhea with LAZ, we also assessed each effect type by diarrhea severity (MSD).

To estimate the association between short-term effect of diarrhea and linear growth in the following month, we used repeated measures linear regression models with generalized estimating equations to account for within-person correlation of repeated measurements on the infant throughout the first year. Diarrhea variables (total diarrhea or MSD episodes in a given month) had a 1-month lag effect to reflect the biologic relationship between diarrhea and linear growth. For example, diarrhea occurring between month one and two were considered exposures relevant to LAZ outcome at three months. Models were adjusted for infant age in months (continuous using restricted cubic splines), LAZ at the start of the monthly interval (continuous), household crowding ($\geq 2$ vs $< 2$ persons per room), exclusive breastfeeding, and maternal education ($>$ primary vs $\leq$ primary education).

To estimate the cumulative burden of diarrhea in the first 11 months on LAZ at 12 months, we used linear regression models adjusted for LAZ at birth, household crowding, exclusive breastfeeding for the first three months of life (75th percentile for exclusive breastfeeding duration), and maternal education. To reduce potential bias in estimates of cumulative diarrhea due to missing visits during the first 11 months, we used multivariable Normal regression

models with Markov chain Monte Carlo method to impute missing values (m = 10 imputations) of the number of diarrhea episodes for missed visits [34]. Length measurements at 12 months were not imputed and thus visits missing LAZ at 12 months were not included in the final analysis.

We used repeated measures linear regression models with generalized estimating equations with an interaction term to assess any difference in the short-term association between diarrhea and linear growth based on the timing of infant diarrhea in the first 1–6 months of age and 7–12 months of age. Diarrhea variables (total diarrhea or MSD episodes in a given month) had a 1-month lag effect. Models included the diarrhea variable of interest, age (an indicator of first or second 6 months of life), an interaction term between diarrhea and age, LAZ at the start of the monthly interval, household crowding exclusive breastfeeding, and maternal education.

The original cohort study and secondary analyses were approved by the Kenyatta National Hospital Ethics and Research Committee and the University of Washington Institutional Review Board. All statistical tests used 2-sided p-values and alpha of 0.05 to determine statistical significance. Participants with missing confounding variables were excluded from relevant analyses. All analyses were conducted in Stata 15.1 (StataCorp, College Station, Texas).

## Results

The parent study enrolled 510 pregnant WLWH and 468 infants with a recorded live birth. Of these 78 were HIV positive at birth or had no HIV negative test after birth, 17 had no regular follow up visit and 1 was missing documentation of infant sex (Fig 1). Characteristics of the 372 remaining HEU infants and their mothers are presented in Table 1. About half (52%) of infants lived in homes with a pit latrine versus a flush toilet and the majority (84%) of infants lived in crowded households ($\geq$2 persons per room). Most (74%) infants were breastfed, and exclusively breastfed for an average of 2.0 months (range 0–7). Less than half (42%) of mothers had more than a primary education. Prior to delivery, mothers' median CD4 count was 450 cells/µl (Inter Quartile Range [IQR]: 316–619) and median HIV VL was 4.7 $\log_{10}$ copies/ml (IQR: 4.1–5.2).

Mean LAZ at birth was 0.31 deviations below the WHO standard (standard deviation [SD]:1.47) and 12% (44) of infants were stunted at birth. By one year of age, mean LAZ among

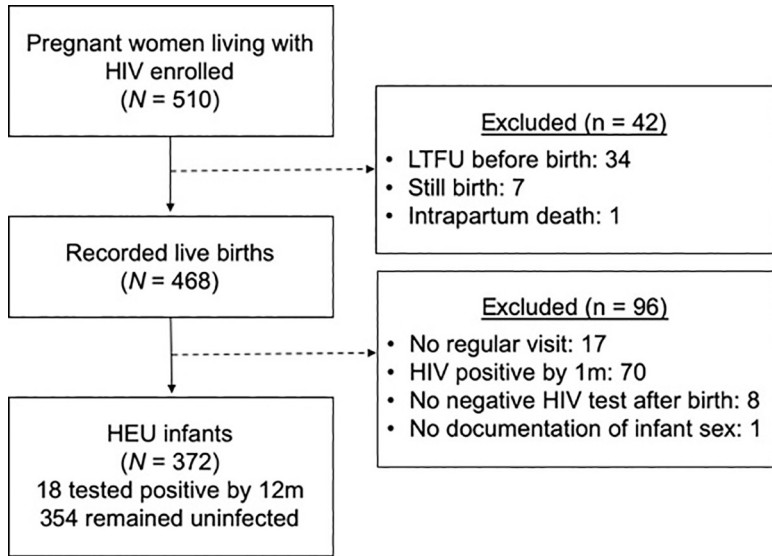

**Fig 1. Flow diagram depicting the section of infants included in the current analysis.**

**Table 1. Selected characteristics of the 372 mothers and HEU infants included in the analysis[1].**

|  | N (%)[2] Mean (min, max) | |
|---|---|---|
| **TOTAL** | 372 | |
| **Home environment factors** | | |
| Pit latrine | 193 | (52) |
| Flush toilet | 179 | (48) |
| ≥ 2 persons/room in house | 314 | (84) |
| < 2 persons/room in house | 55 | (15) |
| **Maternal factors at 32 weeks gestational age** | | |
| Age (years) | 25 | (18, 42) |
| > primary education | 155 | (42) |
| ≤ primary education | 213 | (57) |
| Height (cm) | 161 | (144, 183) |
| MUAC < 23.5 cm | 50 | (13) |
| MUAC ≥ 23.5 cm | 244 | (66) |
| CD4 count < 200 cells/μl | 28 | (8) |
| CD4 count 200–499 cells/μl | 189 | (51) |
| CD4 count ≥ 500 cells/μl | 147 | (40) |
| Log VL ≥ 4 | 266 | (72) |
| Log VL < 4 | 67 | (18) |
| **Infant factors** | | |
| Female | 177 | (48) |
| Male | 195 | (52) |
| Birth LAZ | -0.31 | (-5.49, 4.75) |
| Exclusively breastfed duration (months) | 2.0 | (0, 7) |

[1]Data summary was previously presented elsewhere [35]

[2]Percents may not add to 100% due to missing data

the 268 HEU infants with a 12-month length measurement was -0.97 (SD: 1.2) and 17% (46) of the cohort was stunted. Infants in this cohort experienced 684 episodes of diarrhea and 171 episodes of MSD in 319 infant-years of follow up (mean 2.15 episodes of diarrhea [range: 0–8] and 0.54 episodes of MSD [range 0–4] between birth and 12 months of age). Fig 2 shows mean monthly LAZ among infants who had no diarrhea, at least one episode of diarrhea or at least one episode of MSD in the previous month.

## Short-term effect

MSD and any diarrhea were associated with decreased linear growth in the short term, MSD having a greater magnitude of effect (Table 2). MSD was significantly associated with an average loss of 0.14 (95% Confidence Interval [CI]: -0.24, -0.05) LAZ per MSD episode, while any diarrhea was associated with loss of 0.05 (95% CI: -0.11, 0.01) LAZ per episode and only trended towards statistical significance.

## Cumulative burden

There was no association between LAZ at 12 months of age and cumulative total diarrheal episodes (Table 2). An additional MSD episode in the first 11 months of life was associated with a lower LAZ at 12 months (adjusted mean difference [AD]: -0.08; 95% CI: -0.26, 0.10), although the result was not statistically significant.

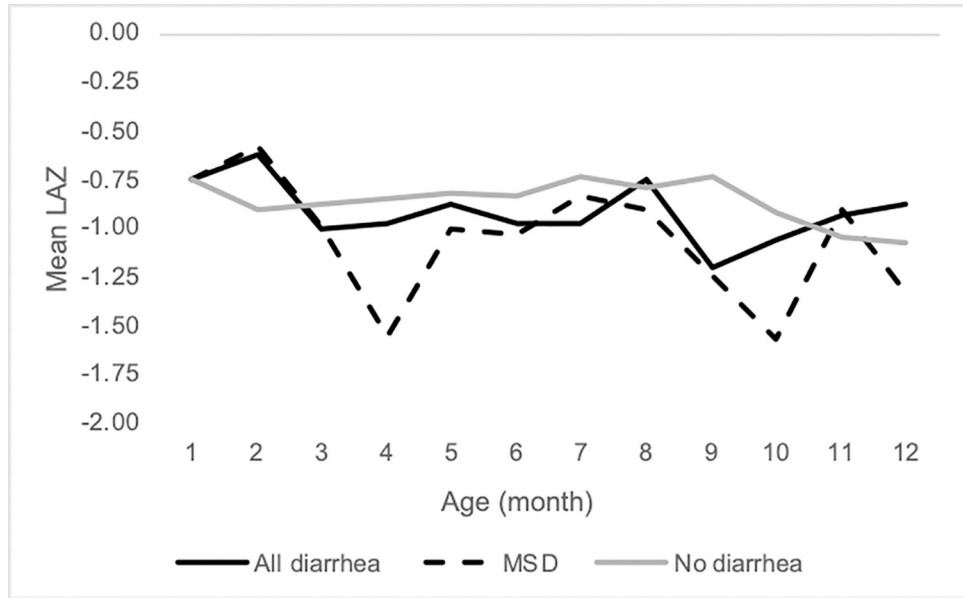

**Fig 2.** Mean length-for-age z-score per month of age among children who had no diarrhea (gray,—), any diarrhea (black,—), or moderate to severe diarrhea (MSD;—-) in the previous month.

### Timing

We did not detect a difference between diarrhea occurring in the first versus the second six months of life in the amount of diarrhea effect on LAZ (Table 3). For both time periods, infants with diarrhea had a lower LAZ in the following month compared to infants without diarrhea. An MSD episode in the first six months was associated with a LAZ decline of -0.14 (95% CI: -0.28, -0.01) in the following month. Similarly, an additional MSD episode in the second six months was associated with lower LAZ (AD: -0.15; 95% CI: -0.29, 0.00). However, there was no difference in effect between the two time periods for all diarrhea (difference in adjusted mean difference [dAD]: -0.04; 95% CI: -0.15, 0.07; p-value interaction: 0.467) nor MSD (dAD: 0.00; 95% CI: -0.21, 0.20; p-value interaction: 0.976).

### Discussion

In response to heterogeneity in the literature regarding the relationship between childhood diarrhea and linear growth, we evaluated the short term and cumulative relationship of

**Table 2. Associations between diarrhea and linear growth.**

|  | Adjusted Difference in LAZ (95% CI)[1] | P-value |
|---|---|---|
| **Difference in LAZ after one month per episode (Short-term)[1]** | | |
| Diarrhea | -0.05 (-0.11, 0.01) | 0.096 |
| MSD | -0.14 (-0.24, -0.05) | 0.003 |
| **Difference in LAZ at 12 months per episode (Cumulative burden)[2]** | | |
| Diarrhea | 0.01 (-0.07, 0.09) | 0.842 |
| MSD | -0.08 (-0.26, 0.10) | 0.368 |

[1]Adjusted for crowding, exclusive breastfeeding in the last 24 hours, and maternal education

[2]Adjusted for crowding, exclusive breastfeeding for at least 3 months, and maternal education

**Table 3. Difference in LAZ after one month per episode of diarrhea occurring in the first 6 months of life versus 7–11 months.**

|  | Adjusted Difference in LAZ (95% CI)[1] | P-value |
|---|---|---|
| **Diarrhea** |  |  |
| 0–6 months | -0.03 (-0.11, 0.06) | 0.561 |
| 7–11 months | -0.07 (-0.14, 0.01) | 0.081 |
| **MSD** |  |  |
| 0–6 months | -0.14 (-0.28, -0.01) | 0.039 |
| 7–11 months | -0.15 (-0.29, 0.00) | 0.054 |

[1]Adjusted for pit latrine, exclusive breastfeeding in the last 24 hours, maternal education, previous months LAZ (diarrhea month)

diarrhea, as well as the influence of the timing and severity of the episode, with subsequent linear growth in a cohort of HEU infants. In our analysis, MSD was associated with short-term (within 1 month) linear growth faltering and this effect did not depend on timing of the MSD episode in an infant's first year of life. We detected a trend for a short-term association between any diarrhea and linear growth. Cumulative burden of any diarrhea or of MSD was not associated with linear growth attainment at 12 months. These findings are consistent with literature showing variable relationships between diarrhea and linear growth and suggest that interventions targeting MSD may have the most impact on growth.

In this study population, the short-term (within 1 month) relationship between diarrhea and linear growth depended on the severity, but not timing of diarrhea. We found that an episode of MSD was associated with a decline of 0.15 standard deviations in LAZ relative to the 0.05 decline found with diarrhea of any severity. Other studies found prolonged and persistent diarrhea [8], and medically attended diarrhea (both MSD and less severe diarrhea (LSD)) to be associated with subsequent linear growth faltering [36] but not community diarrhea [13]. MSD may operate through several mechanisms to influence linear growth. Diarrhea, including MSD, leads to acute weight loss due to both fluid and nutrient loss [6,10,11] and as a child recovers weight following the illness, linear growth may slow [37]. Alternatively, diarrhea may have a direct effect on linear growth through reducing insulin-like growth factor-1 levels [38] or environmental enteric dysfunction [39]. Diarrhea etiologies appear to have a unique relationship with subsequent LAZ [12] and therefore could explain why we observed MSD to have a stronger relationship with growth than any type of diarrhea. Children with MSD (or the characteristics of MSD such as dysentery, hospitalization, and/or dehydration) might be the appropriate population to target for nutritional interventions. Children with more severe diarrhea are also more likely to seek care for diarrhea [40] therefore targeting interventions to health-facility attended diarrhea may result in the greatest improvements in diarrhea-related linear growth faltering.

Repeated diarrhea infections, in combination with inadequate nutrient intake, systemic inflammation, and impaired intestinal absorption are commonly thought to contribute to the cycle of malnutrition and linear growth faltering [9,11]. However, we did not observe such an effect. Our results are consistent with results from the MAL-ED cohort study [12,13], but contrasted with earlier cohort studies finding an association between diarrhea and height attainment at 24 months [10,19]. Cumulative diarrheal burden may only be associated with linear growth past a particular threshold of diarrheal episodes, or the cumulative effect may only become evident after the cessation of breastfeeding and into the second year of life, when the prevalence of stunting peaks [41]. Alternatively, cumulative burden may only have an impact

on linear growth if subsequent episodes are clustered together in time. Some evidence suggests that infants may experience some linear growth decline following a diarrheal episode, but experience linear growth catch-up in the absence of additional diarrheal insults [11]. Therefore, cumulative diarrheal episodes separated by sufficient diarrhea free periods may not have long-term impacts on linear growth.

The infants in this cohort represent a particularly vulnerable group of children. HEU infants may be at increased risk for diarrhea due to in-utero HIV-exposure and subsequent immunosuppression or as a result of increased enteric pathogen exposure from living with HIV-infected household members. Prevention of early diarrhea among HEU infants, through improvements in maternal health, may substantially reduce subsequent linear growth faltering in this vulnerable population [42,43]. Alternatively, there may be other etiologies of linear growth faltering in this vulnerable population that attenuate the contribution of diarrhea to linear growth.

This study had several notable strengths. The use of multiple statistical models allowed us to assess several aspects of the relationship between diarrhea and linear growth. The longitudinal analysis reduces bias due to reverse causality. The cohort included a large number of HEU infants, a representative population of children vulnerable to both diarrhea and growth faltering. In addition, data collection was systematic and allowed for analysis of multiple social, demographic and biological exposures. However, our study had limitations. Data was not originally collected to address hypotheses related to diarrhea and linear growth. Monthly ascertainment of diarrhea morbidity may have under-ascertained diarrhea episodes and may have been limited by recall bias, particularly for less severe episodes [40]. In addition to under-ascertainment of diarrhea, the study population experienced lower rates of linear growth faltering than expected. This could have decreased statistical power, particularly after adjusting for potential confounders, decreasing our ability to detect associations. The parent cohort was recruited prior to widespread maternal ART, cotrimoxazole prophylaxis for HEU infants, and childhood rotavirus vaccination which may influence both diarrhea and linear growth. However, our results provide important mechanistic insights regarding the relationship between diarrhea and linear growth.

In summary MSD was associated with short-term linear growth faltering among HEU infants. Diarrhea severity may play an important role in the relationship between diarrhea and subsequent linear growth. Targeted nutritional interventions for infants with MSD, alongside current prevention and treatment strategies may protect against linear growth faltering in HEU infants. This population represents a particularly vulnerable and growing population in whom reductions in linear growth faltering could have an impact on the global burden of stunting.

## Supporting information

**S1 Checklist.**
(DOCX)

**S1 File.**
(PDF)

## Acknowledgments

We would like to thank the mothers and children who participated in the study, as well as the research personnel, clinic staff, and data management teams in Nairobi Kenya and Seattle, who made this study possible. We thank the University of Nairobi and Kenyatta National

Hospital for support in conducting the parent study and providing administrative and physical infrastructure.

## Author Contributions

**Conceptualization:** Emily L. Deichsel, Grace C. John-Stewart, Judd L. Walson, Patricia B. Pavlinac.

**Data curation:** Dorothy Mbori-Ngacha, Carey Farquhar, Rose Bosire.

**Formal analysis:** Emily L. Deichsel, Barbra A. Richardson, Brandon L. Guthrie.

**Funding acquisition:** Emily L. Deichsel, Grace C. John-Stewart, Carey Farquhar.

**Investigation:** Dorothy Mbori-Ngacha.

**Methodology:** Emily L. Deichsel, Grace C. John-Stewart, Barbra A. Richardson, Brandon L. Guthrie, Patricia B. Pavlinac.

**Project administration:** Grace C. John-Stewart, Dorothy Mbori-Ngacha, Carey Farquhar, Rose Bosire.

**Writing – original draft:** Emily L. Deichsel, Patricia B. Pavlinac.

**Writing – review & editing:** Emily L. Deichsel, Grace C. John-Stewart, Judd L. Walson, Dorothy Mbori-Ngacha, Barbra A. Richardson, Brandon L. Guthrie, Carey Farquhar, Rose Bosire, Patricia B. Pavlinac.

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
