## [Decision Letter · Decision Letter 0]

21 May 2020

PONE-D-20-04710

 Moderate-to-severe diarrhea is associated with short term declines in linear growth among HIV-exposed, uninfected infants

 PLOS ONE

 Dear Dr Deichsel,

 Thank you for submitting your manuscript to PLOS ONE. After careful consideration, we feel that it has merit but does not fully meet PLOS ONE’s publication criteria as it currently stands. Therefore, we invite you to submit a revised version of the manuscript that addresses the points raised during the review process.

Regarding the methods, more details & clarifications are still needed, including of who collected the data and what equipment was used, and was there any kind of inclusion/exclusion criteria considered in collecting data.The discussion section should compare the current results with similar previous studies to draw similarities and distinctions with those other works.The strength and limitations of this study should be added and what this study added to the current knowledge.

For more details, please also review the reviewer's comment below.

 We look forward to receiving your revised manuscript.

 Kind regards,

 Professor Khaled Khatab, PhD.

 Academic Editor

 PLOS ONE

Journal Requirements:

Reviewers' comments:

Reviewer's Responses to Questions

**Comments to the Author**

1. Is the manuscript technically sound, and do the data support the conclusions?

Reviewer #1: Partly

Reviewer #2: Yes

2. Has the statistical analysis been performed appropriately and rigorously? 

Reviewer #1: Yes

Reviewer #2: I Don't Know

3. Have the authors made all data underlying the findings in their manuscript fully available?

Reviewer #1: Yes

Reviewer #2: Yes

4. Is the manuscript presented in an intelligible fashion and written in standard English?

Reviewer #1: Yes

Reviewer #2: Yes

5. Review Comments to the Author

Reviewer #1: I commend the authors for the wonderful job of writing the article not only is scientifically sound but also innovative and a reflection of authors expertise and knowledge in the wider subject areas. However, there are few issues with regards for clarity in the study's objectives, research questions, motivation and other aspects of the manuscript. I have outlined some of my concerns in the attachments accompanying this review.

Reviewer #2: Abstract: It would be worth highlighting in the conclusion that the effect on linear growth was not significant at 12 months.

Introduction: Clearly supports the need for further investigation into this issues and justifies the population group used for this paper.

Methods: Very detailed description of the statistical approaches used in this secondary analysis. Aspects that might be important to include would be details of who collected the data and what equipment was used, particularly with reference to the length as this is highly pertinent to the paper. Similarly, parental reported diarrhoea; how was this standardised as experience suggests that perceptions of what is/isn't diarrhoea vary greatly between individuals.

Results: These are clearly presented and supported by the figures and tables. Table 2 - why is analysis adjusted for at least 3 months exclusive breast feeding when the WHO recommend 6 months?

6. PLOS authors have the option to publish the peer review history of their article (what does this mean?). If published, this will include your full peer review and any attached files.

Reviewer #1: No

Reviewer #2: No

---

## [Author Response · Author response to Decision Letter 0]

17 Jun 2020

Responses to reviewer’s comments on the article titled: “Examining the relationship between diarrhea and linear growth faltering among HIV-exposed, uninfected infants”

Reviewer 1 

1. Comment Under Methodology/Principal Findings, Lines 9-10: Any diarrhea was associated with a trend for LAZ decline (adjusted difference [AD]: 0.05, 95% CI: -0.11, 0.01, p=0.096). How is your claim supported by your results with both CI and p-value showing lack of statistical significance?

Response: We were interpreting a p-value of less than .05 as a trend toward significance but recognize this may not be standard practice. We have removed this sentence from the abstract. 

2. Comment: Abstract Lines10-12: Linear growth outcomes were not predicted by cumulative episodes of diarrheal or MSD, or timing of diarrheal during infancy. Under Author Summary-Lines 5-6: Severity of diarrhea, cumulative burden of episodes, and timing of diarrhea during infancy may influence the relationship between diarrhea and linear growth. Observed seeming contradictory claims!

Response: The first sentence Lines 10-12 summarizes the results from the analysis. The second sentence lines 5-6 in the author summary describes the background and hypothesis for the analysis. We added a semicolon to more clearly link these two sentences.

“ Studies examining the associations between childhood diarrhea and linear growth have yielded differing results; severity of diarrhea, cumulative burden of episodes, and timing of diarrhea during infancy may influence the relationship between diarrhea and linear growth” 

3. Comment: Under Author Summary; Lines 6-8: ‘Children who are born to women living with HIV, who are HIV exposed but uninfected (HEU) have poorer growth and more infectious illnesses compared to infants born to mothers without HIV infection’ Which aspects of the analysis results justifies this conclusion? 

Response: This sentence in the author summary is background information to put our study in context of previous research and is cited in the background section. We acknowledge that this study is unable to assess the relationship between HEU and growth because we did not have a comparison group of children without HIV exposure. 

4. Comment: Under statistical analysis, in para 5 lines 9-10: “…….exclusive breastfeeding, and maternal education (> vs ≤ primary education).” It appears something is missing in the parenthesis. 

Response: Thank you for noticing this. We changed this sentence to say “…….exclusive breastfeeding, and maternal education (> primary vs ≤ primary education).”

5. Comment: The research questions, motivations and objectives of the study are apparent unclear. There is need for understanding of those three vital elements of any research by the readers.

Response: We thank the reviewer for this comment and the opportunity to clarify. Our primary objective of this analysis is to elucidate the relationship between diarrhea and growth within a population of children at high risk of growth faltering. Our research was motivated by heterogeneity in the literature of the effect of diarrhea on growth which makes targeting interventions difficult. We hypothesized that the severity of diarrhea, total burden of diarrhea and the timing of the diarrhea could result in differing relationships between diarrhea and growth. 

We edited the introduction to improve clarity. Research question, motivation and objective are outlined below. 

Research question: “We hypothesize diarrhea severity [8], burden [18,19], and timing of diarrhea [11,20], may explain differences in the diarrhea and linear growth relationship seen in observational and interventional studies.”

Motivation: “Diarrhea disease has long been investigated as a potential cause of stunting with evidence that diarrhea leads to weight loss and eventually linear growth faltering in the absence of sufficient illness-free periods with adequate nutrition to support catch-up growth [6–11]. Results from a robust cohort study found limited effect of childhood diarrhea on linear growth [13] and interventional studies testing treatment and prevention of diarrhea to improve linear growth have had mixed results [14–17].”

Objective: “Utilizing data from a birth cohort of HEU infants from Nairobi, Kenya, we aimed to determine the effect of diarrhea severity, burden, and timing on linear growth.”

6. Comment: Can the outcomes of this study, given its limited scope to Nairobi be a true reflection or generalization of the Kenyan’ situation on the subject matter?

Response: The reviewer brings up an important point. The study population is from an urban setting in Kenya prior to widespread ART during pregnancy or cotrimoxazole prophylaxis for HEU infants. This population may not be generalizable to HEU infants throughout Kenya and we had added language to clarify this point. We do believe these results provide mechanistic insights regarding the relationship between diarrhea and linear growth that may be generalizable to other populations of vulnerable children. 

7. Comment: I am concerned with the study title in its lack of capturing smartly, the central goal of the study. Plausible suggestions: 

i. Effects of Diarrhea Incidence on the linear growth in children: Evidence from Kenya

ii. Effects of Diarrheal on Linear Growth among HIV-exposed uninfected Infants, in Namibia, Kenya

iii. Examining Relationship between diarrheal and linear growth faltering among HIV-exposed, uninfected infants

Response: We appreciate the suggestions for changing the title and have changed the title to be “Examining the relationship between diarrhea and linear growth in Kenyan HIV-exposed, uninfected infants” to use the reviewer’s third suggestion. Thank you! 

8. Comment: What are the distinguishing features of this study from the exiting literature in the research bias? 

Response: We believe the distinguishing features of this study is the assessment of multiple indicators of diarrhea within the same study and their respective associations with linear growth using longitudinal data from a vulnerable population. We added the following sentences in the strengths and limitation sections to clarify. 

“The use of multiple statistical models allowed us to assess several aspects of the relationship between diarrhea and linear growth in the same paper. The longitudinal analysis reduces bias due to reverse causality.”

Reviewer 2 

Comment: Why is analysis adjusted for at least 3 months exclusive breast feeding when the WHO recommend 6 months?

Response: In Table 1 you will note that the median duration of exclusive breastfeeding in this study population was only 2 months. We believe this is in part due a reflection of the rapidly changing guidelines for breastfeeding among women living with HIV during the time period of this study. We wanted to adjustment for potential confounding of excusive breastfeeding on the relationship between cumulative diarrhea and linear growth attainment. Because only 1% of women excusivly breastfed for 6 months we believed it was not a sufficient adjustment for excusivle breastfeeding. We decided to used the 75th percentile of diarrhea duration which was 3 months. We included this detail in the manuscript to improve clarity. The methods section now reads: 

“To estimate the cumulative burden of diarrhea in the first 11 months on LAZ at 12 months, we used linear regression models adjusted for LAZ at birth, household crowding, exclusive breastfeeding for the first three months of life (75th percentile for exclusive breastfeeding duration), and maternal education.”

Editors comments 

Comment: Regarding the methods, more details & clarifications are still needed, including of who collected the data and what equipment was used, and was there any kind of inclusion/exclusion criteria considered in collecting data.

Response: We added information to the methods section clarifying that study staff collected all data using standardized procedures and data collection forms. We also included the use of a length board to measure infant length. Eligibility criteria for pregnant women in the larger study in which this analyses was nested included that they were HIV-seropositive, age ≥18 years, gestation ≤32 weeks, and willing to adhere to scheduled infant blood samplings for at least 1 year. Exclusion criteria for infants in the present analysis are described in the statistical analysis section:

“Singleton or firstborn twin infants with documentation of sex, at least one negative HIV DNA polymerase chain reaction (PCR) test, and at least two recorded length measurements were included in the present analysis.”

Comment: The discussion section should compare the current results with similar previous studies to draw similarities and distinctions with those other works.

Response: We appreciate the editors comment to compare results with additional studies. We have included the following two sentences to further compare results with the other works, for example. 

“Our results are consistent with results from the MAL-ED cohort study [13,14], but contrasted with earlier studies finding an association between diarrhea and height attainment at 24 months”

Comment: The strength and limitations of this study should be added and what this study added to the current knowledge.

Response: Please find a strengths and limitations section in the discussion section on pg 13 and 14.

---

## [Editor Report · Decision Letter 1]

22 Jun 2020

Examining the relationship between diarrhea and linear growth in Kenyan HIV-exposed, uninfected infants

PONE-D-20-04710R1

Dear Dr Deichsel 

We’re pleased to inform you that your manuscript has been judged scientifically suitable for publication and will be formally accepted for publication once it meets all outstanding technical requirements.

Kind regards,

Professor Khaled Khatab, Ph.D.

Academic Editor

PLOS ONE

---

## [Editor Report · Acceptance letter]

14 Jul 2020

PONE-D-20-04710R1 

Examining the relationship between diarrhea and linear growth in Kenyan HIV-exposed, uninfected infants 

Dear Dr. Deichsel:

I'm pleased to inform you that your manuscript has been deemed suitable for publication in PLOS ONE. Congratulations! Your manuscript is now with our production department. 

Kind regards, 

on behalf of

Professor Khaled Khatab 

Academic Editor

PLOS ONE